# Anti-HIV Humoral Response Induced by Different Anti-Idiotype Antibody Formats: An In Silico and In Vivo Approach

**DOI:** 10.3390/ijms25115737

**Published:** 2024-05-24

**Authors:** Valeria Caputo, Ilaria Negri, Louiza Moudoud, Martina Libera, Luigi Bonizzi, Massimo Clementi, Roberta Antonia Diotti

**Affiliations:** 1Pomona Ricerca S.r.l, Via Assarotti 7, 10122 Turin, Italy; 2One Health Unit, Department of Biomedical, Surgical and Dental Sciences, School of Medicine, University of Milan, Via Pascal 36, 20133 Milan, Italy; 3Laboratory of Microbiology and Virology, ‘Vita-Salute’ San Raffaele University, Via Olgettina 58, 20132 Milan, Italy

**Keywords:** minibody, HIV vaccine, anti-idiotype, immunoinformatic, antibody

## Abstract

Despite advancements in vaccinology, there is currently no effective anti-HIV vaccine. One strategy under investigation is based on the identification of epitopes recognized by broadly neutralizing antibodies to include in vaccine preparation. Taking into account the benefits of anti-idiotype molecules and the diverse biological attributes of different antibody formats, our aim was to identify the most immunogenic antibody format. This format could serve as a foundational element for the development of an oligo-polyclonal anti-idiotype vaccine against HIV-1. For our investigation, we anchored our study on an established b12 anti-idiotype, referred to as P1, and proposed four distinct formats: two single chains and two minibodies, both in two different orientations. For a deeper characterization of these molecules, we used immunoinformatic tools and tested them on rabbits. Our studies have revealed that a particular minibody conformation, MbVHVL, emerges as the most promising candidate. It demonstrates a significant binding affinity with b12 and elicits a humoral anti-HIV-1 response in rabbits similar to the Fab format. This study marks the first instance where the minibody format has been shown to provoke a humoral response against a pathogen. Furthermore, this format presents biological advantages over the Fab format, including bivalency and being encoded by a monocistronic gene, making it better suited for the development of RNA-based vaccines.

## 1. Introduction

Since the emergence of HIV infection in 1983, over 85.6 million people have contracted the virus, resulting in an estimated 40.4 million deaths from HIV-1/AIDS, establishing HIV infection as a major global health problem. Currently, about 39 million people worldwide are living with the infection, and, as of 31 December 2022, 29.8 million people have been receiving antiretroviral therapy (ART) [1]. ART is the only available treatment for infected individuals, yet it falls short of resolving the infection; instead, it limits viral replication, slowing down the progression to AIDS. Moreover, ART is a long-term treatment, potentially associated with adverse effects and the emergence of drug-resistant variants [2]. In this scenario, the imperative for developing a safe, effective, and preventive HIV-1 vaccine remains crucial in achieving a durable resolution to the HIV-1/AIDS pandemic [3]. Therefore, the development of effective vaccines is a global priority and a major challenge. Given the viral biological features of HIV-1, the design of vaccine immunogens able to induce a protective immune response is challenging. Considering the importance of neutralizing antibodies in controlling the infection, many studies have focused on identifying the epitope recognized by these antibodies for vaccine development [4,5]. Since Nisonoff et al. proposed the use of anti-idiotype antibodies as an immunogen, candidate anti-idiotypic vaccines have also been developed for HIV-1 infection [6]. In a recent study, Bancroft et al. demonstrated that a panel of anti-idiotype antibodies, designed to specifically recognize the inferred germline version of anti-HIV-1 broadly neutralizing antibody b12 [7,8] (Iglb12), induced the proliferation of transgenic murine B cells expressing the iglb12 heavy chain [9].

In a previous study, we cloned a murine anti-idiotype Fab fragment antibody, P1, obtained from the immunization of mice with antibodies derived from long-term nonprogressor patients [10]. The results show that P1 specifically binds to b12 and induces a neutralizing immune response in rabbits. To enhance the immunogenicity and to bypass the problems connected to the Fab format, we designed two different antibody formats, single chains and minibodies, each with two opposing orientations. We then proceeded to examine their biological characteristics through the utilization of immunoinformatic tools and in vivo experimentation (Figure 1). Our investigation aimed to determine whether these novel formats could elicit an anti-HIV-1 humoral immune response.

## 2. Results

### 2.1. In Silico Analysis of the Different Antibody Formats

Before proceeding with the production and the purification of the different antibody formats, each construct was analyzed using immunoinformatic tools to predict their features as possible immunogens in vaccine preparation. As well defined in the literature, an effective immunogen must exhibit safety, lack toxicity and allergenicity, and be deemed antigenic, qualities that can be explored through in silico methods [11]. The AllerTOP v. 2.0server predicted scFVHVL as a probable allergen (closest protein: NCBI gi number 541736), whereas scFVLVH, MbVLVH, and MbVHVL as a probable nonallergenic (closest protein: UniProtKB accession number Q9NYQ7) (Table 1).

Given the discrepant results between the nonallergenicity and allergenicity of the two single-chain formats, we used two other software programs to clarify it: AlgPred 2.0 and ALLERDET, known for their specificity, sensitivity, and accuracy compared with AllerTop v. 2.0 [12]. Both tools predicted all the constructs as possible allergens (Table 1). In the attempt to modify our molecules to remove allergenic sites, we utilized AlgPred 2.0 to explore potential IgE epitopes. Surprisingly, we did not detect any IgE epitopes during our investigation (Table 1).

ToxinPred2 server showed that all constructs are nontoxin, while VaxiJen 2.0 analysis showed scFVHVL, scFVLVH, Mb VHVL, and Mb VLVH as possible antigens (overall prediction for the protective antigen = 0.6447, 0.6511, 0.6136, and 0.6099, respectively, Table 1).

ProtParam revealed that both scFv P1 formats are unstable proteins, with an estimated molecular weight of ~26.5 kDa, a theoretical pI of 6.3, and a negative value of GRAVY (−0.489). On the contrary, both Mb P1 formats are stable proteins, with an estimated molecular weight of ~39.6 kDa, a theoretical pI of 5.74, and a negative value of GRAVY (−0.560, Table 1).

Given the importance in our study to stimulate a humoral immune response, we used the IEDB server (http://tools.iedb.org/bcell/; Bepipred Linear epitope prediction 2.0, threshold value 0.5) to identify the linear epitope recognized by B cells (Appendix A).

### 2.2. Binding to b12 of the Different Antibody Formats

We proceeded with the cloning and purification of the different constructs. Firstly, all purified antigens were analyzed in ELISA to investigate their capability to bind to b12 Fab fragment, resembling the P1 Fab format, used as a positive control (Figure 2). As expected, apart from negative controls, all antibody formats were able to bind to the b12 Fab fragment. The higher signals were obtained using Fab and Mb VHVL P1, while the lower signal was obtained using scFVLVH P1. The different signals were also highlighted when calculating the affinity of b12/antibody formats of P1 derived by ELISA-based half-maximal effective concentration values when it was possible to calculate (Table 2).

### 2.3. Analysis of Rabbits’ Seroconversion after Immunization with Fab Format

We evaluated the seroconversion of the rabbits using an ELISA on gp120/HIV testing three different serum dilutions. The significance of differences in binding data between pre- and postimmunization samples of the same dilution was calculated using a paired, two-tailed Wilcoxon test (Figure 3). Sera of Fab P1 immunized rabbits were able to bind gp120/HIV as shown by the high signal, and data showed a statistical difference between pre- and postimmunization (*p*-value < 0.1) for each analyzed dilution of sera (Figure 3A). This result was also supported by the negative results obtained using serum samples from rabbits immunized with the negative Fab (Figure 3B). Moreover, the Mann–Whitney U test revealed a significant difference for all tested dilutions between sera of rabbits immunized with Fab P1 and Fab negative (Neg) control (Figure 3C).

### 2.4. Analysis of Rabbits’ Seroconversion after Immunization with scFv Formats

The same experiments were conducted with rabbit sera immunized with scFv formats (Figure 4). The results show that only the scFVHVL P1 format was able to seroconvert the rabbits. A statistical difference between pre- and postimmunization sera (Figure 4B) using the sera diluted to 1:20 was reported, but the reported binding signal was very low (Figure 4B). In fact, we did not report any seroconversion nor statistical difference between the pre and post of mice immunized with scFVLVH (Figure 4C). Also, in this case, the sera of rabbits immunized with the negative control were not able to bind to gp120/HIV (Figure 4A). The statistical analysis among the three groups of rabbits only revealed a significant difference between sera immunized with one of two scFv P1 (scFVHVL P1) with respect to scFv Neg at the dilution of 1:20 (Figure 4D).

### 2.5. Analysis of Rabbits’ Seroconversion after Immunization with Minibody Formats

Immunizations with the different minibody formats all revealed a significant difference between pre- and postimmunization using sera dilution 1:20 and 1:200 (Figure 5). Rabbit sera obtained after immunization with Mb Neg showed a very low binding signal, so the statistical difference cannot be considered (Figure 5A). On the other hand, the highest binding signal was obtained using sera of rabbits immunized with Mb VHVL P1 (Figure 5B). The Mann–Whitney U test within the different rabbit groups revealed that the signal obtained with rabbit sera post immunization with Mb VHVL P1 is significantly different from rabbit sera immunized with the Mb negative control at all test dilutions (Figure 5D).

### 2.6. Statistical Comparison among the Different Formats with Fab P1

We conducted a comparison of the results obtained post immunization with the four different antigens to ascertain the most immunogenic. Employing the Mann–Whitney test, we compared all outcomes with those of Fab P1 immunization, which served as our reference due to their previously reported efficacy [10] (Figure 6). Analysis of the humoral response following immunization with both single-chain formats and Fab revealed a statistically significant difference favoring the Fab format. This trend was consistent across all dilutions (Figure 6). However, a distinct pattern emerged when comparing the immunization outcomes of Mbs with those of Fab. Although the statistical comparison for both types of Mbs did not reach significance, Mb VHVL immunization showed a slightly higher signal on gp120/HIV in comparison with Fab, while Mb VLVH immunization exhibited a lower signal on gp120/HIV compared with Fab (Figure 6).

## 3. Discussion

Despite the introduction of ART in HIV-1 therapy allowing an improvement in the health-related quality of life among HIV-1 patients and a better management of HIV-1 infection, the complete eradication of HIV-1 is unlikely to be reached. Indeed, ART therapy enables the management of HIV infection as a chronic disease, and its administration must be stringent and lifelong with variation in different drug combinations to avoid the onset of drug-resistant viruses. Due to this, the development of a vaccine is necessary to block HIV-1 infection [13].

Considering viral pathobiology, the HIV-1 vaccine should be able to induce a broadly neutralizing immune response to prevent infection of new host cells and an early T cell response to prevent the establishment of a viral reservoir. In the quest to develop a vaccine capable of triggering the production of such antibodies, the identification of epitopes recognized by neutralizing antibodies is crucial. In a previous study, we investigated the long-term nonprogressors’ sera to identify anti-idiotype Fab fragments that could be instrumental in the constitution of a vaccine preparation able to induce an anti-gp120/HIV-1 humoral response. We identified two murine anti-idiotypic Fab fragment formats (P1 and P2) able to induce an anti-HIV-1 humoral response in rabbits, and sera derived from P1 immunized rabbits exhibited the capacity to neutralize an HIV pseudovirus (P1) [10]. Moreover, we also demonstrated that these two Fab fragments serve as the anti-idiotype antibodies of b12, a human broadly neutralizing antibody previously described in the literature [7,8].

To enhance the efficacy of P1 in eliciting an anti-HIV immune response, we explored two antibody formats as immunogens, each with two different orientations, resulting in a total of four different molecules.

To design these molecules, we first eliminated the portion of the antibody that is not useful to induce an anti-gp120 humoral response. Indeed, scFv size is essentially reduced to the variable region (VH and VL) of the antigen-binding part of the antibodies. In this format, the VH and VL are joined together by a flexible polypeptide linker that prevents dissociation and facilitates their pairing, allowing the original antibody binding specificity to be retained in a single polypeptide. For the design of scFv P1, we decided to use 18 amino acids, which guarantees the production of a soluble protein useful for purification, as described by Whitlow et al. [14]. The possibility of using a low-molecular-weight molecule as an immunogen has already been described in the literature; in fact, molecules with the same molecular weight of scFv were used for the immunization of mice against *Trichinella spiralis* [15] and in cattle against *Rhipichepalus appendiculatus* [16].

The second antibody format is based on a study that investigated the efficacy of the b12 minibody format gene transfer to female genital epithelial cells for protection against HIV-1 infection [17]. The minibody format allows one to overcome some limitations inherent in monovalent antibody-derived molecules, such as the Fab and scFv formats. To design the minibody P1, we added a hinge region to the scFvs to obtain a flexible protein [18] and the CH3 portion of the constant region of a murine immunoglobulin (as P1). For the murine CH3 sequence, we used the MAK33 sequence, which guarantees the homodimerization of the protein produced in a prokaryotic system [19], increasing the functional affinity and avidity of this molecule.

Thanks to the improvement that in silico analysis reached in the last years, we predicted that all constructs could be considered as nontoxins and possible antigens.

The data analysis of allergenicity was, however, ambiguous. Two out of three of the used software predicted the constructs as possible allergens, but no IgE epitopes were defined using AlgPred 2.0. This ambiguity may stem from variations in the algorithms underlying the tools used. Additionally, the predicted allergenicity might be attributed to the murine origin of our molecules. Interestingly, analysis of the light and heavy chains sequence of P1 revealed allergenic potential, despite our previous study not indicating any allergenic properties. Physicochemical analysis revealed that both orientations of the minibody format are stable, a very important feature for a protein intended for vaccine preparation [11]. Moreover, all constructs exhibited a negative GRAVY value, indicating their solubility and hydrophilicity nature. Considering the capacity of P1 to induce anti-HIV antibodies [10], we predicted the linear epitope recognized by B cells across all constructs. Interestingly, all the CDRs except CDR2 from both variable regions were identified as epitopes (Appendix A). Due to the unknown structure of our constructs, it is not possible to predict conformational epitopes recognized by B cells.

We proceeded to examine these formats in vitro, evaluating their capacity to bind b12 and subsequently testing sera from rabbits immunized with each antibody format. As anticipated, due to homodimerization, minibodies yielded the strongest signal, with noteworthy differences observed between different orientations (VH-VL and vice versa) of the same antibody format. Notably, the VH-VL orientation exhibited the most favorable results. Similar trends were noted for single-chain formats, consistent with the existing literature, particularly for the multiepitope vaccine, where rearranging the order of epitopes could significantly impact outcomes [11]. This observation was further supported by analyzing sera from immunized rabbits, revealing that employing the minibody with the VH-VL orientation as an immunogen elicited a higher humoral response against gp120/HIV-1 than other formats. The better performance of the minibody format compared with scFv may be attributed to the prolonged half-life of the immunogen, as previously observed with a gp41 fused to IgG1 Fc [20].

Moreover, the interesting data obtained from the immunization with a specific conformation of minibody underscore how even subtle alterations in molecular structure can impact the strength and efficacy of vaccine preparations.

Interestingly, despite the allergenicity prediction defining our molecules as allergens, no allergenic reactions were observed for all immunogens.

The purpose of this paper was to identify the most immunogenic antigen for subsequent characterization. We compared the results of immunization with the four new antibody formats with those with Fab P1. We observed a statistically significant difference favoring the Fab format in terms of humoral response for both single-chain immunizations. While there was a preference for the Fab format over the MbVLVH, the discrepancy was not statistically significant. Contrary to this, in the case of MbVHVL immunization, although the difference between Fab and MbVHVL immunizations was not statistically significant, it is worth noting that a better humoral response was observed following MbVHVL immunization (dilution 1:20 and 1:200).

Our results are the first to demonstrate the minibody format’s potential as an immunogen for eliciting an immune response against a virus. This breakthrough marks a pivotal step towards establishing a polyclonal anti-idiotype minibody vaccine. Compared with conventional anti-idiotype approaches, the minibody format offers several advantages: it is bivalent like a full-sized antibody yet compact in size, and it is monocistronic, both features facilitating the development of mRNA-based vaccines [21]. Furthermore, in contrast to typical anti-idiotype molecules, which may interact with the same target as the pathogen, potentially leading to autoimmune diseases as proposed by Plotz et al. [22], the minibody format studied here is unlikely to induce such adverse effects. Firstly, it is derived from murine antibodies, and secondly, this format lacks the immunoglobulin portion recognized by the immune system [23].

## 4. Materials and Methods

### 4.1. Design and Analysis of P1 scFv and P1 Minibody Using Immunoinformatic Tools

For the design of the P1 single-chain variable fragment format (scFv), the variable region of the light chain (VL) and the variable region of the heavy chain (VH) of P1 coding genes were tethered together with an 18-amino-acid linker. Two different scFvs were designed with opposite orientations of the two variable regions: (i) VH presents at the 5′ terminal (scFVHVL P1); or (ii) VL presents at the 5′ terminal (scFVLVH P1).

As described in a previous article, the minibody was designed by adding a hinge and the CH3 genes to the scFv encoding genes [10]. As for scFv, we designed the same two orientations for the minibody format as well.

AllerTop v. 2.0 (https://www.ddg-pharmfac.net/AllerTOP/ [24], last accessed 26 February 2024), AlgPred 2.0 (https://webs.iiitd.edu.in/raghava/algpred2/batch.html, with a threshold value of 0.3 and a hybrid Machine Learning Technique [25], last accessed 29 February 2024), and ALLERDET (http://allerdet.frangram.com [12], last accessed 29 February 2024) were used to investigate the allergenicity of the constructs. AlgPred 2.0 was also used for the determination of IgE epitopes. Antigenicity and toxicity were evaluated using VaxiJen v2.0 (https://www.ddg-pharmfac.net/vaxijen/VaxiJen/VaxiJen.html, target organism: virus and threshold value 0.4 [26], last accessed 29 February 2024) and ToxinPred2 (https://webs.iiitd.edu.in/raghava/toxinpred2/batch.html, threshold value 0.6 and hybrid Machine Learning Technique [27], last accessed 29 February 2024), respectively.

Physicochemical parameters, such as instability index and grade average of hydropathicity (GRAVY), were analyzed using the ExPaSy ProtParam tool (http://web.expasy.org/protparam/ [28], last accessed 29 February 2024).

The linear epitope of the designed proteins was predicted using the Antigen Sequence Properties tool provided by Immune Epitope DataBase & Tools (IEDB, http://tools.iedb.org/bcell/; Bepipred Linear epitope prediction 2.0, threshold value 0.5, last accessed 29 February 2024).

### 4.2. Cloning and Purification of scFV P1 and Minibody P1

The synthesis of scFvs and minibody encoding genes was performed by GenScript, and the genes were later cloned in a previously described vector [29] using the restriction sites for SacI and SpeI enzymes. For each antibody format, a negative control was designed using variable regions of a murine monoclonal antibody that does not bind to gp120/HIV-1 or other viral proteins. scFvs and P1 minibodies were purified by immune affinity chromatography through Capto^TM^ L resin (Merck, Rahway, NJ, USA). Briefly, bacterial cultures were induced with IPTG (final concentration 1 mM) and were centrifuged after overnight growth at 30 °C. The pellets were resuspended in PBS and then sonicated. Disrupted bacteria were centrifuged, and the supernatants were filtered and passed through the resin column following the manufacturer’s instructions. The purity of the proteins was evaluated through SDS-PAGE. The binding to the b12 antibody was evaluated in ELISA, coating 100 ng/well of each antibody format.

### 4.3. Rabbit Immunization and Sera Characterization

Rabbit immunization and sera collections were performed by Davids Biotechnologie GmbH (Regensburg, Germany). The ability of the different antibody formats to induce an anti-gp120 immune response was evaluated using a modified protocol described in our previous study. Briefly, eight groups of five 5-week-old female rabbits WT New Zealand each were formed as follows: group A, animals immunized with Fab P1 as positive control; group B, animals immunized with Fab unable to bind to b12 (Fab Neg) as negative control; group C, animals immunized with scFVHVL P1; group D, animals immunized with scFVLVH P1; group E, animals immunized with scFv unable to bind to b12 (scFv Neg); group F, animals immunized with MbVHVL P1; group G, animals immunized with MbVLVH P1; and group H, animals immunized with Mb unable to bind to b12 (Mb Neg). The animals were immunized every two weeks (five immunizations) with 120 µg of each antigen resuspended in Addavax, following the manufacturer’s instructions. The sera collected before and after this immunization schedule were analyzed in ELISA at different dilutions (undiluted, 1:20, 1:200) using a commercial monomeric gp120 YU2. Briefly, ELISA plates were coated with 100 ng of recombinant gp120/YU2 and incubated overnight at 4 °C. The plates were blocked with PBS/BSA1% for 1 h at 37 °C, and 40 µL of the different sera were added to the wells and incubated for 1 h at 37 °C. After that, plates were washed five times with PBS/Tween20-0.1%, and the bind to the antigen was revealed with peroxidase-conjugated antirabbit immunoglobulin serum, which was used following the manufacturer’s instructions. After a final wash, 40 µL of TMB substrate was added, and the optical density of 450 nm of the plates was analyzed.

### 4.4. Statistical Analysis

Graphpad Prism v5.0 was used for all statistical analyses. The significance of differences in binding data between pre- and postimmunization samples of the same group was calculated using a paired, two-tailed Wilcoxon test. *p*-value < 0.1 was considered significant due to the reduced number of analyzed samples for each group (five rabbits for each experimental condition), while the significance of differences in binding data between groups was calculated using an unpaired, two-tailed Mann–Whitney U test.

## 5. Patent

International patent application pending (pub. Number WO/2023/100069).

## Figures and Tables

**Figure 1 ijms-25-05737-f001:**
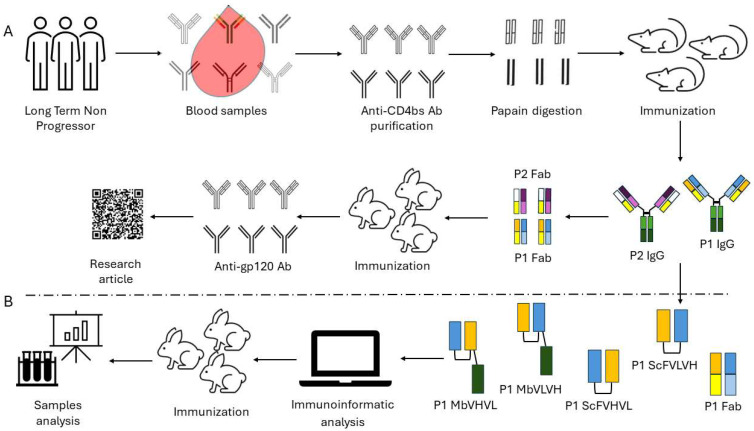
Workflow followed for the construction and assay of the different anti-idiotype antibody formats: (**A**) Workflow followed in our previous work starting from long-term nonprogressors to the characterization of the anti-idiotype antibody P1. (**B**) Workflow employed to design and test the new antibody formats. The design of the different antigens is summarized following a color code scheme.

**Figure 2 ijms-25-05737-f002:**
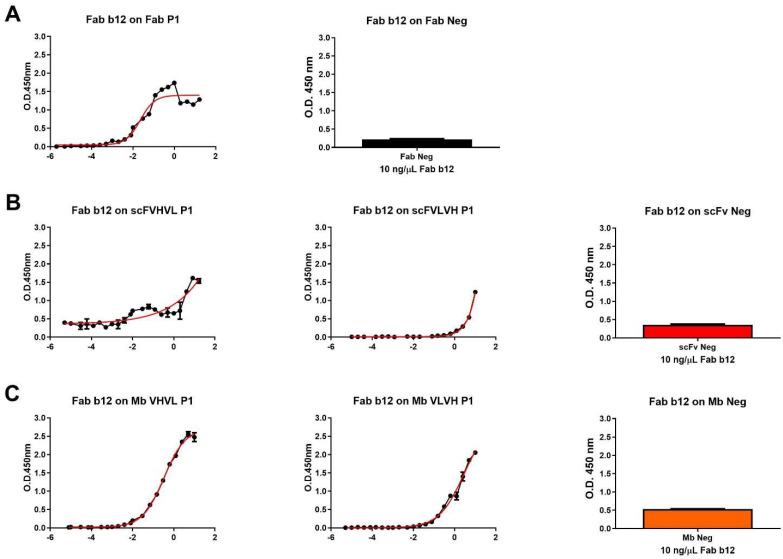
Dose–response curve of b12 Fab on different P1 formats and negative control. (**A**) Dose–response curve of Fab b12 on Fab P1 format (left), used as positive control, and signal obtained using a single concentration of b12 Fab (10 ng/µL) on Fab neg (right). (**B**) Dose–response curve of Fab b12 on scFVHVL P1 (left), scFVLVH P1 (middle), and the signal obtained using a single concentration of b12 Fab (10 ng/µL) on scFv Neg (right). (**C**) Dose–response curve of Fab b12 on Mb VHVL P1 (left), Mb VLVH P1 (middle), and signal obtained using a single concentration of b12 Fab (10 ng/µL) on Mb Neg (right).

**Figure 3 ijms-25-05737-f003:**
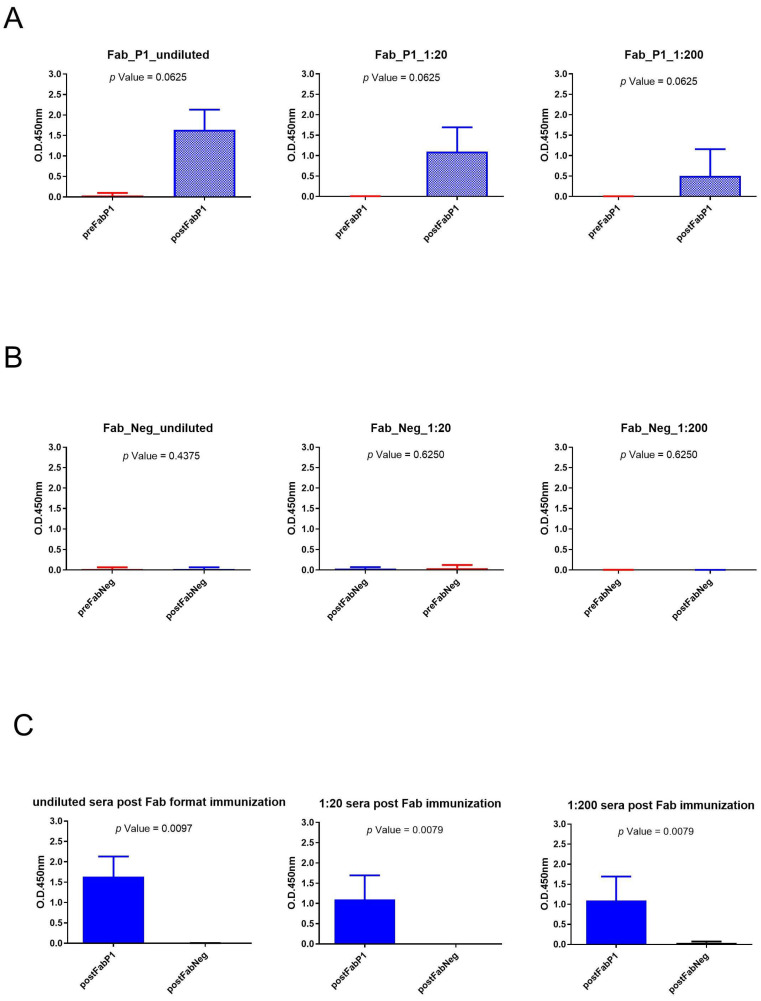
Analysis of Fab immunization sera across various dilutions: (**A**) Wilcoxon test analysis on sera signals obtained using Fab P1 as immunogen on gp120/HIV; (**B**) Wilcoxon test analysis on sera signals obtained using Fab Neg on gp120/HIV. (**C**) Mann–Whitney U test analysis between Fab P1 and Fab Neg data results.

**Figure 4 ijms-25-05737-f004:**
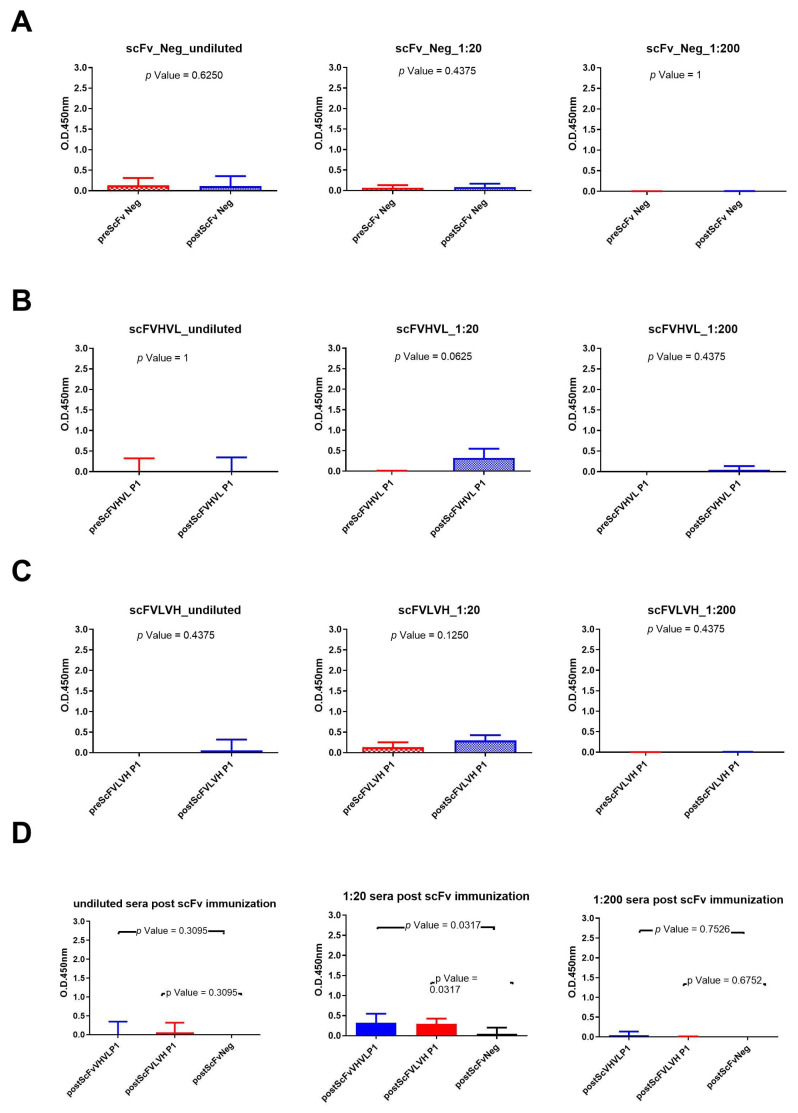
Analysis of scFvs immunization sera across various dilutions: (**A**) Wilcoxon test analysis on sera signals obtained using scFv Neg as immunogen on gp120/HIV; (**B**) Wilcoxon test analysis on sera signals obtained using scFVHVL P1on gp120/HIV; (**C**) Wilcoxon test analysis on sera signals obtained using scFVLVH P1on gp120/HIV. (**D**) Mann–Whitney U test analysis between scFVHVL P1-scFv Neg and scFVLVH P1-scFv Neg data results.

**Figure 5 ijms-25-05737-f005:**
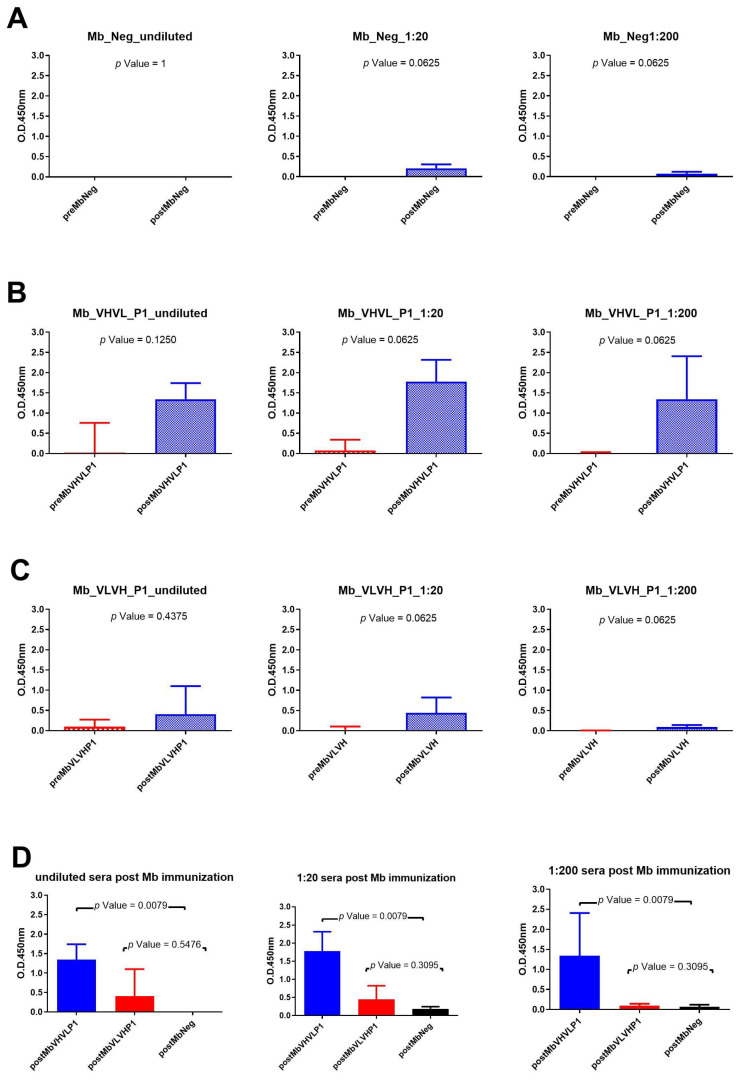
Analysis of Mb immunization sera across various dilutions: (**A**) Wilcoxon test analysis on sera signals obtained using Mb Neg as immunogen on gp120/HIV; (**B**) Wilcoxon test analysis on sera signals obtained using Mb VHVL P1on gp120/HIV; (**C**) Wilcoxon test analysis on sera signals obtained using Mb VLVH P1on gp120/HIV. (**D**) Mann—Whitney U test analysis between Mb VHVL P1-Mb Neg and Mb VLVH P1-Mb Neg data results.

**Figure 6 ijms-25-05737-f006:**
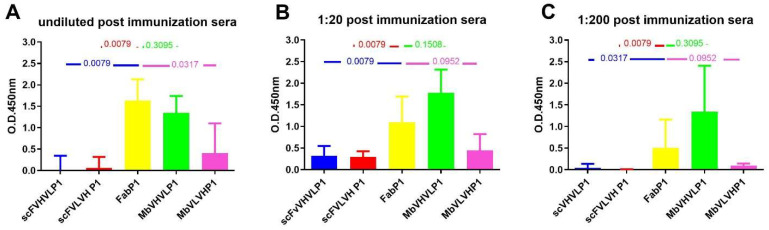
Mann–Whitney U test analysis comparing data obtained from the different formats to Fab P1. The graphs show the statistical assessment of signals on gp120/HIV derived from postimmunization sera, comparing different antibody formats with Fab P1 immunization. The analyses are categorized based on various dilutions: (**A**) undiluted sera; (**B**) 1:20; and (**C**) 1:200. *p*-value is reported in each graph.

**Table 1 ijms-25-05737-t001:** Results of the in silico analysis of the four constructs.

	Allergenicity				Toxicity	Antigenicity	Physicochemical Parameter		
	AllerTOP v. 2.0	AlgPred 2.0	Allerdet	IgE Epitopes (AlgPred 2.0)	ToxinPred 2.0	VaxiJen 2.0	ProtParam		
							Molecular Weight (Da)	pI	GRAVY
**scF VHVL P1**	Allergen	Allergen	Allergen	-	nontoxin	0.6447	26,588.38	6.3	−0.489
**scF VLVH P1**	Nonallergen	Allergen	Allergen	-	nontoxin	0.6511			
**Mb VHVL P1**	Nonallergen	Allergen	Allergen	-	nontoxin	0.6136	33,656.95	5.74	−0.560
**Mb VLVH P1**	Nonallergen	Allergen	Allergen	-	nontoxin	0.6099			

**Table 2 ijms-25-05737-t002:** The half-maximal effective concentration values for each antibody format were calculated using statistic software (GraphPad Prism v5.0) and converted in molarity. Due to the ambiguous dose–response curve obtained with the two different orientations of the scFvs, it was not possible to define the half-maximal effective concentration values.

Antibody Formats	ELISA-Based Half-Maximal Effective Concentration Values (M)
Fab P1	0.02268
scFVHVL P1	ambiguous
scFVLVH P1	ambiguous
Mb VHVL P1	0.33
Mb VLVH P1	2.8

## Data Availability

The data presented in this study are available within the article.

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
