# Peer review of "Anti-HIV Humoral Response Induced by Different Anti-Idiotype Antibody Formats: An In Silico and In Vivo Approach"

_ijms, 2024, doi:10.3390/ijms25115737_

Round 1

Reviewer 1 Report

Comments and Suggestions for Authors

In this study, the authors designed two forms of antibody against human immunodeficiency virus type 1 (HIV-1) broadly neutralizing antibody B12, and further explored the biological properties of the molecules by both immune-bioinformatics and sera analyses.

However, there are several deficiencies in the manuscript:

Majors

1. The efficacy to the candidates in the current form of MS is preliminary. Since the goal is supposed to elicit humoral responses to mimic broadly neutralizing Ab b12, authors should test if the rabbits immunized with candidates could show any neutralization against living HIV challenge or pseudo-viruses. The binding test to gp120 could only certify the specificity to viral antigen.

2. The titers of candidate immunization sera were too low to be promising in further investigation. The dilution of 1:200 only shows the ability in binding but the affinities and the expected biological functions could hardly to be optimistic.

Minors,

1. Returning to the essence of immunology, the authors did not explain the source of B cell epitopes, and the relationship between the role of epitopes, antibodies and B lymphocytes was not clearly explained, which is important for the design of vaccines. In addition, regarding the structural design of the four anti-idiotype antibodies, it may be more convenient for readers to comprehend with a pattern diagram here.

2. The authors used a variety of bioinformatics tools to explore the properties of the molecules and tested allergenicity by a variety of methods. How to explain the difference between the results of different algorithms, and is there any candidate considered possible allergens excluded from the scope of vaccine design.

3. The picture in the article is not clear and the quality of figures is poor. It is highly recommended to improve the quality of the images.

Comments on the Quality of English Language

Minor editing of English language required.

Reviewer 2 Report

Comments and Suggestions for Authors

The manuscript entitled: “ANTI-HIV HUMORAL RESPONSE INDUCED BY DIFFERENT ANTI-IDIOTYPE ANTIBODY FORMATS: AN IN SILICO AND IN VIVO APPROACH” deals with a very interesting topic of great importance: finding a proper immunogen for developing a vaccine against HIV. In general, it is a well written manuscript. However, there are some major and minor concerns that need to be addressed before the manuscript is reconsidered for publication.

·     The major concern is the authors’ choice of significance level of 0.1 due to the number of rabbits (n=5)/group. Although the theory of statistics may justify this choice under certain circumstances (eg pilot studies or unusual set of data), a significance level of 0.05 is most widely used in studies including similar number of experimental animals. More specifically, in the overwhelming majority of publications regarding antibody testing in rabbits the sample size varies from 4-6 rabbits/group and the significance level is set at 0.05. The authors should either increase the number of rabbits/group or set the significance level at 0.5 and reconsider if the differences between pre- and post-immunization sera are statistically significant and then discuss their findings on this basis.

·   The authors used their antibodies although in silico analysis mostly showed that they were probable allergens. In discussion (line 185) they claim that no IgE epitopes were identified. This finding, although shown in table 1, should also be written in paragraph 2.1.1. Moreover, since they decided to use them despite the “negative” in silico predictions they should write that no allergic reactions were observed in immunized rabbits.

·    Please explain better the color code in Supplementary tables. Moreover, place the headings for each table before and not after the table.

·      Figure legends should more explicative, explain the experiment and stand by themselves so that the reader does not have to read the text to understand the figure.

·      Figures are very small and with low analysis that makes it very difficult to read them. Please replace them with ones with better analysis and larger fonts.

·  Figures referring to the sera of the rabbit immunized with the same antibody format (5A, B and C should be part of Figures 2, 3 and 4 respectively so that in the text the reader does not jump from figure 2 to figure 5 and then back to 3 and 5 again etc.

·      In line 107 the authors refer to statistical significance in figures 3B and 3C. No statistically significant results are shown in figure 3C.

·      Negative OD values in figures 2, 3 and 4 have no biological meaning and should be shown as zero.

·      How do the authors explain that in most cases there is no statistically significant difference between pre- and post-immunization undiluted sera?

·   The authors should check again their references section, there are mistakes. E.g. Reference 13 is the same with ref 9, Reference 11 in line 271 does not seem to be appropriate.

Reviewer 3 Report

Comments and Suggestions for Authors

Comments for the Authors

ijms-2942453

ANTI-HIV HUMORAL RESPONSE INDUCED BY DIFFERENT ANTI-IDIOTYPE ANTIBODY

FORMATS: AN IN SILICO AND IN VIVO APPROACH

Authors: Valeria Caputo *, Ilaria Negri, Louiza Moudoud, Martina Libera, Luigi Bonizzi, Massimo Clementi, Roberta Antonia Diotti

In this manuscript, Caputo and colleagues attempted to determine the best antibody format based on the previously described b12 anti-idiotype, P1. They claimed that, according to their in silico analysis, the minibody format is a better candidate antigen than the single-chain format. Moreover, using sera collected from the rabbit animal model (eight groups of five 5-week-old female rabbits WT New Zealand) with a period of five times immunizations, the authors indicated that a specific minibody confirmation demonstrated a strong b12 binding signal and could elicit a robust humoral response against HIV-1 infection in rabbits.

Major comments:

1. Four candidates, ScFVHVL, ScFVLVH, Mb VHVL, and Mb VLVH have been released and tested in this work; however, a lack of a conclusion in the main text, stating the optimal choice among these four antibody formats. Can the authors provide a summary of the results from these four antibody formats and describe their pros and cons in detail?

2. The number of CD4+ T cells is one of the critical indicators for HIV-1 infections. Can the authors provide the measurement of the alteration of CD4+ T cell counts in rabbits and verify whether these four antibody candidates enable CD4+ T cell reconstitution?

3.  Please improve the quality of all figures. The labels shown in each panel are difficult to read.

Minor comment:

1. Following the flow of the content, Fig 5 could be placed in advance as Fig 3.

Round 2

Reviewer 1 Report

Comments and Suggestions for Authors

It still needs further evaluation to make the study sound and rigorous.. The current form is too preliminary.

Comments on the Quality of English Language

It looks fine.

Author Response

Thank you for your feedback. Given the scope of the article we believe that the results are clear and sufficient; however, as you suggested, we will for sure conduct further experiments in the future.

Reviewer 2 Report

Comments and Suggestions for Authors

The authors have improved their manuscript and have responded adequately to my comments.

Two minor comments:

Line 131: Do authors mean Fig 3C? There is no Fig 3D.

Line 405. The authors did remove reference 11 which was not relevant with the text, however they need to add the proper one.

Author Response

We have made the corrections highlighted in green.

Reviewer 3 Report

Comments and Suggestions for Authors

Comments for the Authors

ijms-2942453 (revised version)

ANTI-HIV HUMORAL RESPONSE INDUCED BY DIFFERENT ANTI-IDIOTYPE ANTIBODY FORMATS: AN IN SILICO AND IN VIVO APPROACH

Authors: Valeria Caputo *, Ilaria Negri, Louiza Moudoud, Martina Libera, Luigi Bonizzi, Massimo Clementi, Roberta Antonia Diotti

Thank you for providing a revised version of this manuscript. The manuscript is now better structural and clear. The Figure 6 is appreciable. 

Below please find two minor comments:

  1. In Line 131 I am wondering if the author attempted to cite Fig. 3C rather than Fig. 3D. I cannot find Fig. 3D in this revised manuscript.

  2. Fig. 4C is never mentioned in this version of this manuscript. The authors could consider describing these panels or removing them if not necessary.

Author Response

(The authors gave the same response as above.)
